# Singular Spectrum Analysis for Background Initialization with Spatio-Temporal RGB Color Channel Data

**DOI:** 10.3390/e23121644

**Published:** 2021-12-07

**Authors:** Huy D. Le, Tuyen Ngoc Le, Jing-Wein Wang, Yu-Shan Liang

**Affiliations:** 1Department of Electronic Engineering, National Kaohsiung University of Science and Technology, Kaohsiung 80778, Taiwan; lehuydip@gmail.com (H.D.L.); ysliang@nkust.edu.tw (Y.-S.L.); 2Department of Electronic Engineering, Ming Chi University of Technology, New Taipei City 24301, Taiwan; 3Institute of Photonics Engineering, National Kaohsiung University of Science and Technology, Kaohsiung 80778, Taiwan

**Keywords:** background initialization, separation of foreground and background, singular spectrum analysis, spatio-temporal data

## Abstract

In video processing, background initialization aims to obtain a scene without foreground objects. Recently, the background initialization problem has attracted the attention of researchers because of its real-world applications, such as video segmentation, computational photography, video surveillance, etc. However, the background initialization problem is still challenging because of the complex variations in illumination, intermittent motion, camera jitter, shadow, etc. This paper proposes a novel and effective background initialization method using singular spectrum analysis. Firstly, we extract the video’s color frames and split them into RGB color channels. Next, RGB color channels of the video are saved as color channel spatio-temporal data. After decomposing the color channel spatio-temporal data by singular spectrum analysis, we obtain the stable and dynamic components using different eigentriple groups. Our study indicates that the stable component contains a background image and the dynamic component includes the foreground image. Finally, the color background image is reconstructed by merging RGB color channel images obtained by reshaping the stable component data. Experimental results on the public scene background initialization databases show that our proposed method achieves a good color background image compared with state-of-the-art methods.

## 1. Introduction

Scene background initialization is a basic low-level process in video-processing applications, such as video segmentation [1], video compression [2], computational photography [3], and video surveillance [4,5] (e.g., tracking, counting). The background initialization is also known as background estimation, background reconstruction, and background generation. The task of background initialization can be described as follows: given a video, we need to construct a model that describes the clear background image despite the continued presence of moving objects. The background image may be valid for the entire video or updated in time if the background configuration changes due to illumination change or the displacement of background objects.

Figure 1a shows frames from the *HighwayII* sequence of the scene background initialization (SBI) database [6]. There is an appearance of moving objects in each frame, particularly cars. These frames are the input data of the background initialization model as described in Figure 1b. Using the proposed background initialization model, we can eliminate the appearance of moving objects to obtain a clean background, which is also known as the closest-to-ground-truth background, as shown in Figure 1c.

During the past two decades, many methods [1,7,8,9,10,11,12,13,14,15,16,17,18,19,20,21,22,23,24,25] were proposed for the background initialization task. In general, these techniques can be classified into four main categories: pixel-based methods [1,7,8,9,10,11], iterative-based methods [12,13,14,15], low-rank/sparse data separation methods [16,17,18,19,20,21,22], and deep learning-based methods [23,24,25,26,27].

The first subcategory includes pixel-based methods where each pixel is processed individually over time. Chiu et al. [1] achieved the background by clustering the pixels. Pixels obtained from each location along its time axis are clustered according to their intensity variations. The pixel corresponding to the cluster that has a maximum probability more significant than a time-varying threshold is extracted as a background pixel. Maddalena and Petrosino [7] used a temporal median to compute the background pixel as the mean of the pixels at the same position across all the image sequences. The most well-known method is a mixture of Gaussians (MoG) proposed by Stauffer [9]. The background is modeled probabilistically at each pixel location by fitting MoG to the observed pixel valued in a recent temporal window. MoG decides whether each pixel is classified as background or foreground. More recently, Laugraud et al. [10] presented a method called LaBGen, which combined a pixel-wise temporal median filter and a patch-selection mechanism based on motion detection. In each frame, a background subtraction algorithm determines whether each pixel in the video belongs to the foreground or background. Tian et al. [11] introduced the block-level background modeling (BBM) algorithm to obtain video-coding background components. The BBM algorithm uses the residual gradient as the temporal information to distinguish the background blocks. BBM is used to consider the boundary difference, and the pixel smoothness process is handled using a weighted average of pixel temporal value.

The second subcategory includes iterative-based methods [12,13,14,15]. These methods usually consist of two stages. In the first stage, these methods detect static regions considered reference backgrounds. The background model is iteratively completed in the second stage based on suitable spatial consistency criteria. Hsiao and Leou [12] performed background initialization and foreground segmentation tasks based on motion estimation and computation of the correlation coefficient. Each block of the current frame is classified into four categories: background, still object, illumination change, and moving entity to exploit for the background updating phase. The static blocks, such as “background” and “illumination change”, are selected as the reference, and the remaining blocks are suitably used for the iterative completion of the background model. In [13], Torre and Black applied robust principal component analysis (RPCA) for separating the background and foreground to detect the outlier from video or image data. Firstly, the number of bases that preserve 55% of data energy is calculated using standard PCA. Then based on the obtained number of bases, RPCA is used for minimizing the vital energy function until convergence to receive the weight matrix. Finally, the weight matrix is used to detect outliers. Reitberger and Sauer [14] proposed a background-determining model based on an iterative singular value decomposition via singular vectors spanning a subspace of the image space. The method has a fast processing speed and can be applied in real-time applications. But it has difficulty handling challenges, such as intermittent motion. Recently, based on long-term background stable and short-term foreground changes of scenes, Chen et al. [15] adopted a Bayesian framework to classify the background and foreground. 

The third subcategory includes low-rank/sparse data separation methods. The background information is considered low-rank information, and the remainder of the data represents both noises and moving objects. One of the first attempts to initialize the background in this subcategory was introduced by Candes et al. [16]. They perfectly separated a given video into a low-rank matrix and a sparse matrix by solving a very convenient convex program called principal component pursuit (PCP). However, PCP has several disadvantages for real-world videos, such as its time consumption and computational complexity. To overcome the limitations of the PCP method, many studies were proposed, such as Javed et al. [17] and Zhou et al. [18], which work well in specific environments. Ye et al. [19] presented a motion-assisted matrix restoration (MAMR) model for background-foreground separation of a video. In the MAMR model, the sparse matrix contains the foreground objects, and the low-rank matrix includes the background. A dense motion field is calculated and mapped into a weighting matrix for each frame, which indicates the likelihood that each pixel belongs to the background. In [20], Grosek and Kutz introduced the video dynamic mode decomposition (DMD) method for foreground and background separation. The DMD method decomposes video data into different dynamic modes, which are associated with Fourier frequencies. The frequencies near the origin do not change from frame to frame. Thus they are considered background components, and the terms with Fourier frequencies bounded away from the origin are foreground components. In [21], non-negative matrix factorization (NMF) was used to approximate a non-negative matrix *A* to a product of two non-negative, low-rank factor matrices *W* and *H*, where *W* contains background components and *H* contains foreground components. More recently, Kajo et al. [22] introduced a spatio-temporal, slice-based, singular value decomposition (SVD) method by organizing videos, such as tensors and seeks, to sparse them into different components. Each of these components, namely the moving object and the background, is represented by a few distinct significant eigenvalues. However, this proposal can be time-consuming to process over an ample space. Besides, it still has some limitations in the complex scenes, such as illumination variation, short video, and clutter.

The fourth subcategory includes deep learning-based methods [23,24,25,26,27]. These methods used the effectiveness of the deep learning model to automatically learn the background model. Ramirez-Quintana and Chacon-Murguia [23], based on self-organizing maps (SOMs) and cellular neural networks (CNNs), proposed a self-adaptive system named SOM-CNN. This system includes two neural network architectures called retinotopic SOM (RESOM) and neighbor threshold CNN (NTCNN) for video and motion analysis. The system can work with typical and complex scenarios in real time. Zhao et al. [24] proposed a background modeling method called the stacked multilayer self-organizing map background model (SMSOM-BM). This model can learn the background model of challenging scenarios and automatically determine most network parameters by considering every pixel and spatial consistency at each layer. Halfaoui et al. [25] proposed a CNN-based method to estimate the background component. This method is effective for challenges, such as dynamic backgrounds, illumination variation, and clusters. Yang et al. [26] proposed a deep neural network for background modeling. First, they used the temporal encoding to sample multiple frames from original sequential images with variable intervals, then they used a fully convolutional network to extract temporal and spatial information from frames. In the work by Gregorio et al. [27], the authors introduced a background initialization approach by weightless neural network. Each pixel is allied to an artificial weightless neural network that learns more frequently. This method is useful for processing long-term and live videos.

In the real world, background initialization still faces many challenges, such as lighting changes, the foreground occupying most of the frames, the automatic adjustment of the video camera, and objects moving heterogeneously (sometimes stationary, sometimes moving). To address these issues, we propose a novel method belonging to the low-rank/sparse data separation method named background initialization with singular spectrum analysis (BISSA). Firstly, the input image sequence is reorganized into a spatio-temporal data type useful for background–foreground separation tasks. Secondly, an adaptive background initialization algorithm for image sequences based on the SSA is proposed. Finally, to evaluate the effectiveness of our method, we compare our approach with some of the state-of-the-art techniques by doing experiments on a SBI [6] database. The experiment results show that our proposed method is more accurate and easier to apply in real-world applications.

The rest of the paper is organized as follows: Section 2 describes an overview of the SSA algorithm. Section 3 presents our proposed method. Finally, experimental results and discussion are summarized in Section 4, while conclusions and future work are represented in Section 5.

## 2. Singular Spectrum Analysis

In recent years, singular spectrum analysis (SSA) [28,29,30] has emerged as a powerful non-parametric tool to apply for analyzing and predicting time series data. This method aims to decompose the input data into a sum of different meaningful components, where these components can be grouped and merged based on their common properties to compose subsequent components. These grouped components indicate different groups of features of the original time series data. Currently, many researchers apply SSA in different areas, such as biomedical diagnostic tests [31], climatology [32], economics [33,34], signal processing [35], etc. A flowchart of SSA, consisting of the substages of decomposition and reconstruction, is shown in Figure 2.

As can be seen in Figure 2, the basic SSA algorithm consists of two isolated stages: decomposition and reconstruction stages. In the first stage, embedding and singular value decomposition steps are applied for the decomposition. In the last stage, eigentriple grouping and diagonal averaging steps are used to reconstruct the time series. For example, given non-zero time series $X=\left(f_{1},f_{2},\ldots ,f_{K}\right)$ of length K, W is denoted as the window length and 1<W<K; L=K−W+1. The SSA algorithm is described below:


**Stage 1: Decomposition**



**Step 1: Embedding**


Embedding is a standard procedure in time series analysis. Embedding can be regarded as a mapping that transfers a one-dimensional time series into a multidimensional series. By selecting a large window size, more information about the basis pattern of the time series is captured. Constructing the trajectory matrix F of the original time series X, which is a matrix of size W×L, gives:(1)$$F={\left(\begin{array}{ccccc} f_{1} & f_{2} & f_{3} & \ldots & f_{L}\\ f_{2} & f_{3} & f_{4} & \ldots & f_{L+1}\\ f_{3} & f_{4} & f_{5} & \ldots & f_{L+2}\\ \vdots & \vdots & \vdots & \ddots & \vdots \\ f_{W} & f_{W+1} & f_{W+2} & \ldots & f_{K} \end{array}\right)}_{W\times L},$$
where rows and columns of F are subseries of the original time series.


**Step 2: Singular value decomposition (SVD)**


This step computes the SVD of the trajectory matrix F sized W×L. By using SVD, matrix F can be decomposed into the product of three matrices: an orthogonal matrix U of size *W* × *r*, a diagonal matrix Σ of size r×r, and the transpose of another orthogonal matrix V of size r×L, where r is the rank of matrix F. In general, the SVD of trajectory matrix F can be written as:(2)$$F=U\Sigma V^{T}=\overset{r}{\underset{i=1}{\sum }}u_{i}\sigma_{i}v_{i}^{T},$$
where $U=\left[u_{1},u_{2},u_{3},\ldots ,u_{r}\right]$ and $V=\left[v_{1},v_{2},v_{3},\ldots ,v_{r}\right]$ are the column-orthonormal matrices, respectively (i.e., $U^{T}U=I$ and $V^{T}V=I$), and $\Sigma =diag\left(\sigma_{1},\sigma_{2},\sigma_{3},\ldots ,\sigma_{r}\right)$ is a diagonal matrix containing the singular values (SVs) of F, where SVs are arranged in the descending order $\left(\sigma_{1}\geq \sigma_{2}\geq \sigma_{3}\geq \ldots \geq \sigma_{r}>0\right)$. The matrices $F_{i}=u_{i}\sigma_{i}v_{i}^{T}$ are called elementary matrices: they have rank-1. The collection $\left(u_{i},\sigma_{i},v_{i}\right)$ is called i-*th eigentriple* of matrix F.


**Stage 2: Reconstruction**



**Step 3: Eigentriple grouping**


This step can be used to analyze and determine the physical behavior of each component in the time series data. The purpose of the eigentriple grouping is to gather data based on their common properties. The different matrices of rank-1 acquired from applying the SVD of trajectory matrix F can be selected and gathered together. In that way, correctly clustered groups reflect other original time series data criteria. The grouping procedure separates the set of r eigentriples into m (m≤r) distinct subsets, and they are expressed as $F_{G_{j}}=\{F_{G_{1}},F_{G_{2}},\ldots ,F_{G_{m}}\},$ where each $F_{G_{j}}$ contains several $F_{i}$ and presents as:(3)$$F_{G_{j}}=\left[\begin{array}{ccccc} f_{11,j} & f_{21,j} & f_{31,j} & \ldots & f_{L1,j}\\ f_{12,j} & f_{22,j} & f_{32,j} & \ldots & f_{L2,j}\\ f_{13,j} & f_{23,j} & f_{33,j} & \ldots & f_{L3,j}\\ \vdots & \vdots & \vdots & \vdots & \vdots \\ f_{1W,j} & f_{2\left(W+1\right),j} & f_{3\left(W+1\right),j} & \ldots & f_{LK,j} \end{array}\right].$$

The progress of selecting the sets $F_{G_{1}},F_{G_{2}},F_{G_{3}},\ldots ,F_{G_{m}}$ is called *eigentriple grouping*. 


**Step 4: Diagonal averaging**


The final step is to perform the diagonal averaging on the matrices $F_{G_{j}}$
where j=1,2,3,…,m. This step converts grouped matrices $F_{G_{j}}$ into a one-dimensional original time series via the diagonal averaging method. In particular, where $F_{G_{j}}$ is a trajectory matrix grouped in step 3, the element ${\tilde{f}}_{kj},\;k=1,2,\ldots ,K$ of time series data $S_{j}$ is computed as the average of all elements on the minor diagonal *k*th of matrix $F_{G_{j}}$, which can be expressed as:(4)$${\tilde{f}}_{kj}=\frac{1}{k}\underset{\begin{array}{c} x+y=k+1\\ 1\leq x,\;y\leq k \end{array}}{\sum }f_{xy,j},\;k=1,2,\ldots ,K.$$

The result of the reconstructed trajectory matrix along the diagonal averaging process is time series data of length K represented by:(5)$$S_{j}=\left\{{\tilde{f}}_{1j},{\tilde{f}}_{2j},{\tilde{f}}_{3j},\ldots ,{\tilde{f}}_{Kj}\right\}.$$

## 3. Background Initialization Using Singular Spectrum Analysis

Generally, background–foreground separation can be regarded as a matrix separation problem [16,36,37,38,39,40]. We can separate a video into two group components, one component that contains stable information and the remaining component that holds dynamic information. Constructing these components can be based on an eigentriple or a group of eigentriples. The background data (almost stable and highly correlated between frames) is contained in the static component, and the dynamic component usually represents the foreground data (moving objects or noise). The matrix separation problem can unify in a more general framework formulated as follows [16,36,37,38,39,40]:(6)$$X=S+\varepsilon_{D},$$
where X is the input video data information, matrix S indicates the stable component, and $\varepsilon_{D}$ represents the dynamic component, respectively. These components are achieved by reconstructing one or a group of eigentriples of the trajectory matrix F. As a result, the stable and dynamic components are calculated as:(7)$$S=\overset{\tau }{\underset{i=1}{\sum }}u_{i}\sigma_{i}v_{i}^{T},$$
(8)$$\varepsilon_{D}=\overset{r}{\underset{j=\tau +1}{\sum }}u_{j}\sigma_{j}v_{j}^{T},$$
where 1≤τ≤r and r is the rank of F. In this study, video X is stored in three matrices as spatio-temporal data $M^{\left(C\right)}$. Trajectory matrices $F^{\left(C\right)}$ are constructed based on these spatio-temporal data matrices, where C∈{R, G, B} represents *R*, *G*, and *B* color channels. More details on how to construct video X as spatio-temporal data used as the input data for our background initialization system are introduced in the following subsections.

### 3.1. Storing a Video as Spatio-Temporal Data

A fundamental problem in mathematics is how to arrange data, through which they reveal the most critical information. By organizing the correct given data, we can solve our problem. In this section, we introduce a way of rearranging input video data to solve the problem of separating the background and moving objects. Spatio-temporal data [40] is a data type that contains both space and time characteristics of the original data. Spatial refers to space and temporal relates to time. Spatio-temporal data analysis is discovering patterns and knowledge from spatio-temporal data. A video can be considered a dynamic system with evolving frames, where each frame presents the system’s state. In this study, by flattening the color frames of a video as columns of matrices, we obtain spatio-temporal data.

A grayscale video is three-dimensional (3D) input data, which is the frame height (m), width (n), and time (k) with k frames of the video, as shown in Figure 3a. By reshaping each frame into a column of size 1×a of a matrix of size k×a (where a=m×n), as shown in Figure 3b, we obtain the spatio-temporal data matrix. In this matrix, the correlation between pixels located at the same neighboring position between adjacent frames is preserved over time. Additionally, the video is mapped from 3D space into two-dimensional (2D) space, thereby reducing the complex computing.

Without loss of generality, X is assumed to be an original color video consisting of *k* frames with a resolution of *a*. To display multichannel images in the RGB space, 24 bits with 8 bits for each color channel is used. Firstly, each color frame is separated into three color channels, namely *R*, *G*, and *B*. Secondly, we flatten each color channel frame to one vector and arrange the vector side by side to form a spatio-temporal data matrix, called the color channel spatio-temporal matrix. Finally, we obtain three spatio-temporal data matrices corresponding to the three color channels. Based on the color frame, we construct three spatio-temporal data matrices. The process to flatten video’s frames to the color channel spatio-temporal data matrices is summarized in Algorithm 1, as follows:
**Algorithm 1.** Construct the Three-Color Channel Spatio-Temporal Data Matrices of the Video**Input:***X* is a color video consisting of *k* color frames, where each frame has a resolution of *a* = *m* × *n*.**Output:** Three color channel spatio-temporal data matrices,
$$M^{\left(C\right)},\;C\;\in \left\{R,\;G,\;B\right\}$$, of the video.1.*f*_1_, *f*_2_, … , *f_k_*←Extract *k* frames *f_i_* of the video *X*.2.$$f_{i}^{\left(C\right)}$$←Separate frame *f_i_* into RGB color channel images, *i* = 1, 2, 3,.., *k*.3.$$m_{i}^{\left(C\right)}$$←flatten each
$$f_{i}^{\left(C\right)}$$ image into a vector column of size 1 × *a*.4.$$M^{\left(C\right)}$$←Arrange the vector column
$$m_{i}^{\left(C\right)}$$ side by side to form color channel spatio-temporal data matrices.

### 3.2. Singular Spectrum Analysis for Background Initialization

This section presents the central part of our background initialization method using SSA in detail. We introduce how to apply SSA for the background initialization task, given that X is a color video sequence of k frames, where each frame has a resolution of a=m×n. Firstly, by using Algorithm 1, as discussed in Section 3.1, we receive three color channel spatio-temporal data matrices $M^{\left(C\right)}$ of size a×k, C∈{R, G, B} representing the *R*, *G*, or *B* color channel used, which can be written as:(9)$$M^{\left(C\right)}={\left(\begin{array}{ccccc} f_{11}^{\left(C\right)} & f_{12}^{\left(C\right)} & f_{13}^{\left(C\right)} & \ldots & f_{1k}^{\left(C\right)}\\ f_{21}^{\left(C\right)} & f_{22}^{\left(C\right)} & f_{23}^{\left(C\right)} & \ldots & f_{2k}^{\left(C\right)}\\ f_{31}^{\left(C\right)} & f_{32}^{\left(C\right)} & f_{33}^{\left(C\right)} & \ldots & f_{3k}^{\left(C\right)}\\ \vdots & \vdots & \vdots & \ddots & \vdots \\ f_{a1}^{\left(C\right)} & f_{a2}^{\left(C\right)} & f_{a3}^{\left(C\right)} & \ldots & f_{ak}^{\left(C\right)} \end{array}\right)}_{a\times k}.$$

**Embedding:** We construct the trajectory matrices $F^{\left(C\right)}$ based on color channel spatio-temporal data matrices $M^{\left(C\right)}$ by embedding operator T. The dimensions of the matrices $F^{\left(C\right)}$ are determined by two window lengths,$\;W_{a}$ and $W_{k}$, where $1\leq W_{a}\leq a$, $1\leq W_{k}\leq k,$ and $1<W_{a}W_{k}<ak$, then $L_{a}=\left(a-W_{a}+1\right)$ and $L_{k}=\left(k-W_{k}+1\right)$. The input 2D matrix $M^{\left(C\right)}$ is organized into the matrix $F^{\left(C\right)}$ of size $\left(W_{a}W_{k}\times L_{a}L_{k}\right)$ as follows:(10)$$T(M^{\left(C\right)})=F^{\left(C\right)}={\left(\begin{array}{ccccc} T_{1}^{\left(C\right)} & T_{2}^{\left(C\right)} & T_{3}^{\left(C\right)} & \ldots & T_{L_{k}}^{\left(C\right)}\\ T_{2}^{\left(C\right)} & T_{3}^{\left(C\right)} & T_{4}^{\left(C\right)} & \ldots & T_{L_{k+1}}^{\left(C\right)}\\ T_{3}^{\left(C\right)} & T_{4}^{\left(C\right)} & T_{5}^{\left(C\right)} & \ldots & T_{L_{k+2}}^{\left(C\right)}\\ \vdots & \vdots & \vdots & \ddots & \vdots \\ T_{W_{k}}^{\left(C\right)} & T_{W_{k+1}}^{\left(C\right)} & T_{W_{k+2}}^{\left(C\right)} & \ldots & T_{k}^{\left(C\right)} \end{array}\right)}_{W_{a}^{\left(C\right)}W_{k}^{\left(C\right)}\times L_{a}^{\left(C\right)}L_{k}^{\left(C\right)}},$$
where each $T_{i}^{\left(C\right)}$ is a trajectory matrix of size $W_{a}^{\left(C\right)}\times L_{a}^{\left(C\right)}$ composed from the color channel spatio-temporal data matrix $M^{\left(C\right)}$, such as:(11)$$T_{i}^{\left(C\right)}={\left(\begin{array}{ccccc} f_{1i}^{\left(C\right)} & f_{2i}^{\left(C\right)} & f_{3i}^{\left(C\right)} & \ldots & f_{L_{a}i}^{\left(C\right)}\\ f_{2i}^{\left(C\right)} & f_{3i}^{\left(C\right)} & f_{4i}^{\left(C\right)} & \ldots & f_{L_{a+1}i}^{\left(C\right)}\\ f_{3i}^{\left(C\right)} & f_{4i}^{\left(C\right)} & f_{5i}^{\left(C\right)} & \ldots & f_{L_{a+2}i}^{\left(C\right)}\\ \vdots & \vdots & \vdots & \ddots & \vdots \\ f_{W_{a}i}^{\left(C\right)} & f_{W_{a+1}i}^{\left(C\right)} & f_{W_{a+2}i}^{\left(C\right)} & \ldots & f_{ai}^{\left(C\right)} \end{array}\right)}_{W_{a}^{\left(C\right)}\times L_{a}^{\left(C\right)}}.$$

**Decomposition**: We perform SVD on the trajectory matrices $F^{\left(C\right)}$ to obtain sets of the rank-1 matrices.
(12)$$F^{\left(C\right)}=U_{\left(C\right)}\Sigma_{\left(C\right)}V_{\left(C\right)}^{T}=\overset{r_{\left(C\right)}}{\underset{i=1}{\sum }}u_{i}^{\left(C\right)}\sigma_{i}^{\left(C\right)}{\left(v_{i}^{\left(C\right)}\right)}^{T},$$
where $U_{\left(C\right)}=\left[u_{1}^{\left(C\right)},\;u_{2}^{\left(C\right)},\ldots ,u_{r_{\left(C\right)}}^{\left(C\right)}\right]$ and $V_{\left(C\right)}=\left[v_{1}^{\left(C\right)},\;v_{2}^{\left(C\right)},\ldots ,v_{r_{\left(C\right)}}^{\left(C\right)}\right]$ are orthogonal matrices containing singular vectors, $\Sigma_{\left(C\right)}=diag\left(\sigma_{1}^{\left(C\right)},\;\sigma_{2}^{\left(C\right)},\ldots ,\sigma_{r_{\left(C\right)}}^{\left(C\right)}\right)$ contains sorted SVs in a non-increasing order, and $r_{\left(C\right)}$ is the rank of $F^{\left(C\right)}$.

**Grouping**: The rank-one matrices are merged following general criteria; the aggregate of the rank-one matrices acquire the grouped matrices in N (N≤r) groups.
(13)$$F^{\left(C\right)}=F_{G_{1}}{}^{\left(C\right)}+F_{G_{2}}{}^{\left(C\right)}+\ldots +F_{G_{N}}{}^{\left(C\right)},$$
where $F_{G_{m}}^{\left(C\right)}=\overset{N}{\underset{m=1}{\sum }}u_{m}^{\left(C\right)}\sigma_{m}^{\left(C\right)}{\left(v_{m}^{\left(C\right)}\right)}^{T}.$

**Return to the object decomposition**: The grouped matrices are transformed to the form of the input object by performing $T^{-1}$ based on the diagonal averaging method, as described in Equation (4):(14)$${\tilde{F}}_{G_{m}}^{\left(C\right)}=T^{-1}\left(F_{G_{m}}^{\left(C\right)}\right),\;$$
where m=1,2,3,…,N.

### 3.3. Grouping of Eigentriples

This section analyzes and determines the specific meaning of an eigentriple or a group of eigentriples in video data. The first step is to set a window length. The algorithm proposed in this study separates the set of eigentriples into two groups, as described in Equation (6). Both groups reconstruct output data, resulting in two reconstructed output component data for given input data, so we set all window lengths to 2 in this study.

We selected video sequences, namely Board, CaVignal, and IBMtest2, for analysis and observation. This experiment considers window length sizes 2, 4, and 10, respectively. Figure 4 presents the eigenvectors plot from trajectory matrices of the three videos, as we analyzed the data with different window length sizes. As shown in Figure 4, from top to bottom, the blue line represents the first component, and other color lines indicate the remaining components. We can see that the first eigenvector is always a constant over time. The eigenvalue represents the magnitude of the data, and the eigenvector indicates the direction of the data. Therefore, the first eigenvector represents the unchanged data component over time. Those are referred to as stable components (S), representing the background in the video. Because of that reason, we reconstructed the background in this first eigentriple-based video and dynamic component $\left(\varepsilon_{D}\right)$ obtained by remaining eigentriples.

In summary, we split the set of indices {1,2,…,r} into two groups, namely a stable component and a dynamic component. The result of the step is the representation:(15)$$S=u_{1}\sigma_{1}v_{1}^{T},\;$$
(16)$$\varepsilon_{D}=\overset{r}{\underset{i=2}{\sum }}u_{i}\sigma_{i}v_{i}^{T},\;$$
where *r* is the rank of trajectory matrix.

### 3.4. Proposed Method

From the arguments presented above, by using the first eigentriple of color channel spatio-temporal data matrices, we can construct the most effective background image of the given video. The remaining eigentriples are used to construct the foreground. The implementation of the main part of BISSA method can be summarized in Algorithm 2 as follows:
**Algorithm 2.** Initialize Background Using SSA**Input:** Color channel spatio-temporal data matrices $M^{\left(C\right)}$ of video *X*.**Output:** Obtain background and foreground images corresponding to each input color frame.1.Construct trajectory matrix $F^{\left(C\right)}:$
$$F^{\left(C\right)}\leftarrow Equation\left(10\right)\left(M^{\left(C\right)}\right)$$2.Decompose the trajectory matrix $F^{\left(C\right)}$ into a sum of one-rank elementary matrices：
$$F^{\left(C\right)}=U_{\left(C\right)}\Sigma_{\left(C\right)}V_{\left(C\right)}^{T}=\overset{r_{\left(C\right)}}{\underset{i=1}{\sum }}u_{i}^{\left(C\right)}\sigma_{i}^{\left(C\right)}{\left(v_{i}^{\left(C\right)}\right)}^{T},$$ where $\;r_{\left(C\right)}=rank\left(F^{\left(C\right)}\right).$3.Construct background component $S^{\left(C\right)}$ based on the first eigentriple group (i=1):
$$S^{\left(C\right)}=u_{1}^{\left(C\right)}\sigma_{1}^{\left(C\right)}{\left(v_{1}^{\left(C\right)}\right)}^{T}$$4.Construct the foreground component $\varepsilon_{D}^{\left(C\right)}$ based on remaining eigentriple groups:
$$\varepsilon_{D}^{\left(C\right)}=\overset{r_{\left(C\right)}}{\underset{i=2}{\sum }}u_{i}^{\left(C\right)}\sigma_{i}^{\left(C\right)}{\left(v_{i}^{\left(C\right)}\right)}^{T}\;$$5.Perform the diagonal averaging $S^{\left(C\right)}:$
$${\tilde{S}}^{\left(C\right)}\leftarrow Equation\;14\left(S^{\left(C\right)}\right)\;$$6.Perform the diagonal averaging $\varepsilon_{D}^{\left(C\right)}:$
$${\tilde{\varepsilon }}_{D}^{\left(C\right)}\leftarrow Equation\;14\left(\varepsilon_{D}^{\left(C\right)}\right)\;$$7.Reshape the first column of ${\tilde{S}}^{\left(C\right)}$ to matrices sized *m* × *n*:
$${\tilde{S}}^{\left(C\right)}=reshape\left({\tilde{S}}^{\left(C\right)},\;m,n\right)\;$$8.Reshape the columns of ${\tilde{\varepsilon }}_{D}^{\left(C\right)}$ to matrices sized *m* × *n*:
$${\tilde{\varepsilon }}_{D,i}^{\left(C\right)}=reshape\left({\tilde{\varepsilon }}_{D,i}^{\left(C\right)},\;m,n\right),$$ where i={1,2,3,…,k}.9.Merge three colors channels of ${\tilde{S}}^{\left(C\right)}$ to receive the background image of the video:
$$S_{bg}=merge\left({\tilde{S}}^{\left(C\right)}\right)$$10.Merge three colors channels of ${\tilde{\varepsilon }}_{D}^{\left(C\right)}$ to receive the foreground image $\varepsilon_{bg,\;i}$ corresponding to $i^{th}$ frame:
$$\varepsilon_{bg,\;i}=merge\left({\tilde{\varepsilon }}_{D,i}^{\left(C\right)}\right)$$

Given X is a video consisting of k color frames, where each frame has a resolution of a=m×n, after applying Algorithm 1, the three color channel spatio-temporal data matrices corresponding to three color channels are obtained. Each color channel spatio-temporal data matrix contains k columns and a rows. Each column corresponds to one frame, and each row contains k pixel values of the same pixel position in the video. By applying Algorithm 2 on the three color channel spatio-temporal data matrices separately, we process and find the relevance of all frames over time. In our proposed method, we use the eigentriples of the color channel spatio-temporal data matrix to construct two groups of matrices. The first group is constructed by using only the first eigentriple, which contains the background information of the video. The second group is built by using the remaining eigentriples, which include the foreground information. To receive the desired background images, we reshape each column with the first reconstruction matrix to a matrix of size m×n to obtain the chosen background images. The color background image is obtained by merging the three color channels. Moreover, k-achieved color background images are the same; thereby, only the first column of the first matrix is used to reconstruct the background image to save processing time. Experimental results that support our arguments are discussed in more detail in the next section.

## 4. Experimental Results and Discussion

In this section, to indicate the effectiveness of our proposed method, experiments for the background initialization problem are conducted on the most popular benchmark database named the SBI [6] database. We also compare our background initialization performance with some state-of-art background initialization, such as median [7], RPCA [13], dynamic mode decomposition (DMD) [20], non-negative matrix factorization (NMF) [21], and background estimation by WiSARD (Wilkes, Stonham and Aleksander Recognition Device) [27]. Finally, to assess the accuracy of the obtained background images against the ground truth images, we use several measurement metrics, such as structural similarity index (SSIM) [41], feature similarity index for image quality assessment (FSIM) [42], peak-signal-to-noise ratio (PSNR) [43], average gray-level error (AGE), and percentage of error pixels (pEPs) [44].

### 4.1. SBI Database

This database was introduced by L. Maddalena at the workshop on scene background modeling and initialization in 2016. The SBI database [6] consists of 14 different image sequences, namely *Board*, *Candela_m1.10*, *CAVIAR1*, *CAVIAR2*, *CaVignal*, *Foliage*, *Hall&Monitor*, *HighwayI*, *HighwayII*, *HumanBody2*, *IBMtest2*, *People&Foliage*, *Snellen*, and *Toscana*, as shown in Figure 5a. These sequences are composed of 6 to 740 frames, and their dimensions vary from 144×144 to 800×600. Each image sequence is accompanied by a ground truth background image, as shown in Figure 5b. SBI was designed to evaluate existing and future background initialization algorithms. The image sequences in the SBI database are intended for different challenges in background initialization tasks, such as camera jitter and shadows challenge, intermittent motion challenge, clutter challenge, very short challenge, etc.

In the SBI database, the first one is the clutter challenge, where the objects appear almost to cover the entire background, such as the *Board*, *People&Foliage*, *Foliage*, and *Snellen* sequences. The *HighwayI* and *HighwayII* sequences have many cars that are constantly moving. These sequences include challenges, such as shadows and camera jitter. The *Candela_m1.10* sequence presents a scenario where a man appears with his bag and leaves the scene with nothing. The *CaVignal* sequence is challenging because the man appears and retains position in more than 60% of the frames before leaving. Some sequences include the challenges of intermittent motion, such as *Candela_m1.10*, *CAVIAR1*, *CAVIAR2*, *CaVignal*, *Hall&Monitor*, and *People&Foliage*. The other sequences, *HumanBody2* and *IBMtest2*, contain basic challenges. Finally, the *Toscana* sequence consists of only 6 frames, which presents a short video challenge. Our goal is to propose a background estimation method to obtain the factual background of a given video.

### 4.2. Evaluation and Result

Our paper proposes an efficient method for the background initialization task. As discussed in Section 3, given X is a color video sequence of *k* frames $f_{i},\;i=1,\;2,\;\ldots ,\;k$ with a resolution of a=m×n, the background initialization algorithm based on SSA is proposed. Firstly, color frames are split into three color channels, $f_{i}^{\left(C\right)}$, C ∈{R, G, B} representing the *R*, *G*, or *B* color channels, then flattened to a vector column of color channel spatio-temporal data matrices $M^{\left(C\right)}$. These matrices contain both space and time characteristic information of the original video. In $M^{\left(C\right)}$ matrices, the correlation between pixels located at the same position between adjacent frames is preserved over time. Next, $M^{\left(C\right)}$ matrices are decomposed by SSA to find the eigentriples for constructing the stable and dynamic components. The stable component containing the background information is computed by grouping the first eigentriple, and the remaining eigentriples construct the dynamic component. Finally, by reshaping an arbitrary column of ${\tilde{S}}^{\left(C\right)}$ to matrices of size m×n, then merging the three color channels, we receive a corresponded background image of the video. Similarly, by reshaping the columns of ${\tilde{\varepsilon }}_{D}^{\left(C\right)}$ to matrices of size m×n, then merging the three color channels, we obtain a sequence of foreground images corresponding to each video frame. 

Figure 6 displays achieved background and foreground images corresponding to the frames in *HighwayI*, *IBMtest2*, *CAVIAR2*, and *HighwayII* sequences by using our proposed method. The background images presented in the second row and the foreground images corresponding to the frames are illustrated in the third row. These videos contain several challenges, such as intermittent motion, shadows, camera jitter, and basic. Using our proposed method, for each video containing k frames, we can obtain k background image and k foreground image. This study focused on the background initialization task; however, we also obtained impressive foreground results, as shown in the third row of Figure 6. As can be seen, all moving objects are eliminated from the original frame’s image. However, all the background images are the same, as shown in the second row of Figure 6. Therefore, we only need to construct a video’s background image by reshaping one column of the stable component to an image to reduce processing time. 

Our proposed algorithm can obtain background and foreground images of a video. However, we focus on handling the background initialization in this paper. To show the greater effectiveness of BISSA, we compare the proposed method with several existing methods, such as median [7], RPCA [13], DMD [20], NMF [21], and BEWiS [27]. 

Figure 7 shows the ground truth background images and obtained background images using different methods of 14 different video sequences in the SBI dataset. The *Toscana* sequence in the SBI database includes only six frames that represent the challenge of very short videos. Therefore the convergence criterion of the *Toscana* sequence is not met in RPCA, which means we cannot compute the matrix that contains the weighting of each pixel in the training data. Figure 7a presents ground truth background images of 14 video sequences in the SBI dataset. Figure 7b–g illustrate the background images obtained by using BEWiS, median, RPCA, DMD, NMF, and our proposed method, respectively. As seen, our proposed method obtains a clear background image in most cases, such as the *Board*, *Candela_m1.10*, *CAVIAR1*, *CAVIAR2*, *Hall&Monitor*, *HighwayI*, *HighwayII*, *HumanBody2*, *IBMtest2*, and *Snellen* image sequences. For the *CaVignal* image sequence, the obtained background image is not as expected because the man appears and retains position in more than 60% of the frames before leaving, much like the *People&Foliage* video sequence, in which the result is not expected because the people and trees appear in 338 out of 341 frames. For the *Toscana* video sequence, the results are not as good as expected due to too few frames (only six frames) and the object appears to occupy the majority of the video. In summary, our proposal achieves positive results on the basic challenge, intermittent motion challenge, camera jitter challenge, and shadows challenge, but struggles a little in handling clutter video and a very short video sequence. However, the obtained results are still really good when compared to other methods, such as RPCA, DMD, and NMF.

To assess the accuracy of the obtained background images against the ground truth images, we use five measurement metrics: SSIM [41], FSIM [42], PSNR [43], AGE, and pEPs [44], to measure the similarity between the two images. These measurement metrics are image-to-image metrics measuring the visual correctness of an estimated background image against a ground truth background image. These methods exploit different aspects of image quality evaluation, thus leading to an extensive comprehensive evaluation of the obtained result. Table 1 summarizes the rank of values and preference of these measurement metrics. As can be seen in Table 1, for the SSIM, FSIM, and PSNR measurement metrics, the higher obtained values demonstrate a higher similarity between the two images. On the contrary, for the AGE and pEPs measurement metrics, the lower of the obtained values show a higher similarity between the obtained backgrounds and ground truth images. A summary is presented in Table 1. 

A summary is presented in Table 2 that highlights the best values of the corresponding metrics in bold. As shown in Table 2, the BISSA method gets high performance in most videos when we use the SSIM, FSIM, and PSNR metrics to evaluate. With the pEPs metric, our proposed method gets high performance in most videos except the *Board*, *CaVignal*, and *Snellen* sequences. BEWiS is the best performer in the *Foliage*, but BISSA is still better than the RPCA, DMD, median, and NMF methods. With the AGE metric, our proposed method gets high performance in most videos except *Canlenda_m1.10*, *CAVIAR1*, and *HumanBody2*. However, the BISSA is still better than the remaining methods. When using the PSNR metric to evaluate results, our proposed method also gets high performance in most videos except *Canlenda_m1.10* and *Snellen*. In most videos, the backgrounds obtained by our proposed method are very similar to the ground truths.

## 5. Conclusions

This study proposes an effective background initialization algorithm for image sequences. By storing color frame sequences of the video into color channel spatio-temporal data matrices, we can preserve the correlation between pixels located at the same position between adjacent frames over time. Next, the SSA method was applied to these spatio-temporal data. Then, the stable component is constructed by using the first eigentriple, which is the component that holds the color background image. In addition, encouraging results of the foreground component were obtained based on the remaining eigentriples. The experiment results on the most popular public scene background initialization database demonstrate our proposed method’s effectiveness. The obtained background image is compared to the ground truth background image by the five most common metrics: SSIM, FSIM, PSNR, AGE, and pEPs. The results proved that our study achieved some positive results, especially in dealing with basic challenges, cluster challenges, intermittent motion challenges, camera jitter challenges, and intense shadow challenges. In addition, the results also show that our proposed method achieves a good color background image when compared with state-of-the-art techniques, such as BEWiS, median, RPCA, DMD, and NMF. However, videos recorded with few frames (less than 20 frames) and intermittent object motion challenges (such as *CaVignal* sequence) remain open challenges. Moreover, computing the background from the first eigentriple only obtains a good estimation of the background, but is not optimal to get a reasonable estimate of the foreground. In the future, we will continue to improve our method to achieve better background and accurately detect moving objects in video.

## Figures and Tables

**Figure 1 entropy-23-01644-f001:**
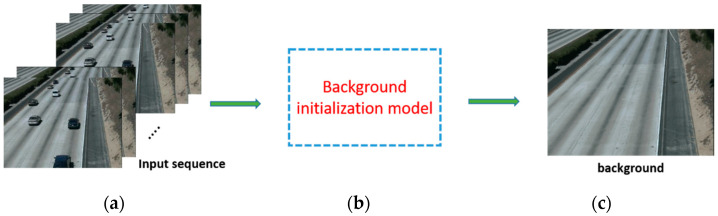
The background initialization task: (**a**) image sequence, (**b**) background initialization model, and (**c**) desired background.

**Figure 2 entropy-23-01644-f002:**
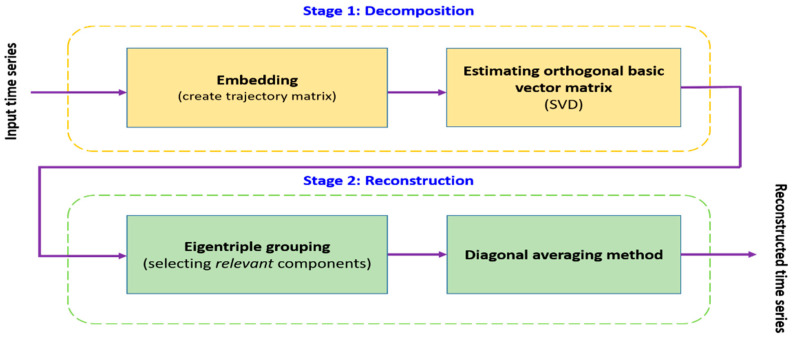
Flowchart of singular spectrum analysis algorithm.

**Figure 3 entropy-23-01644-f003:**
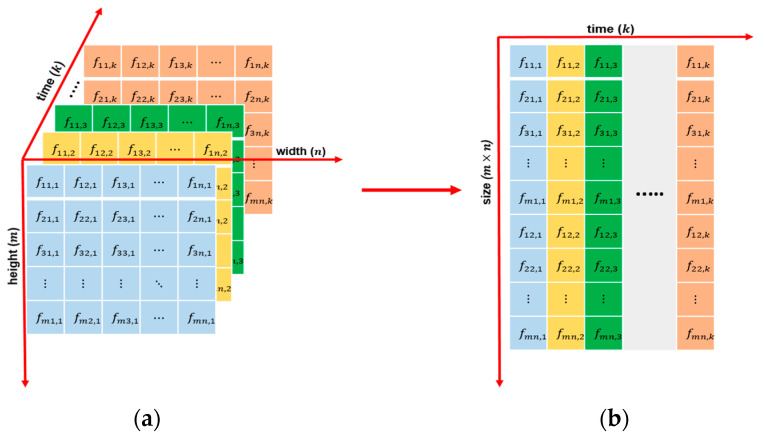
The process of storing a video as a spatio-temporal data matrix: (**a**) a color video sequence of k frames with a resolution of m×n and (**b**) a spatio-temporal data matrix where each column represents one frame of the original video.

**Figure 4 entropy-23-01644-f004:**
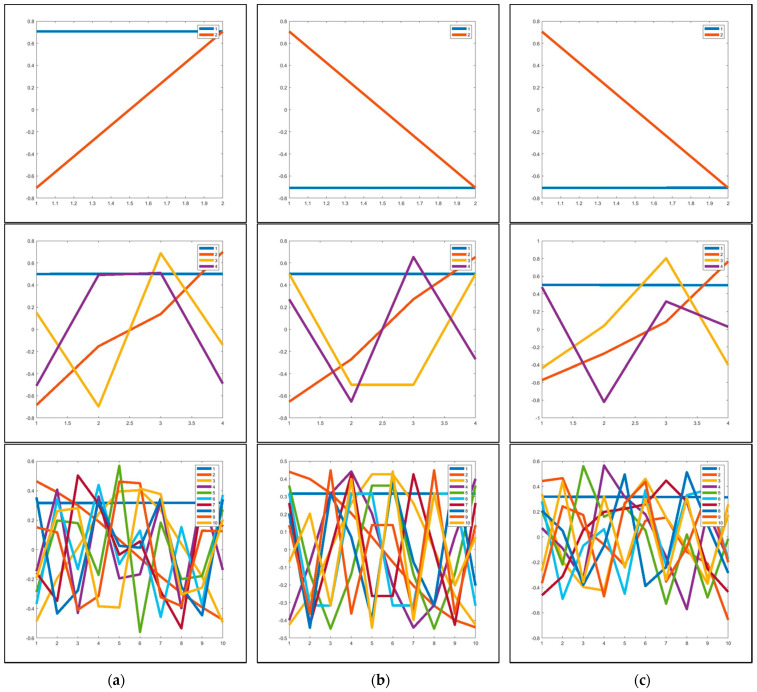
Eigenvectors plot of trajectory matrices for SBI database: (**a**) Board sequence, (**b**) CaVignal sequence, and (**c**) IBMtest2 sequence. The blue line indicates the direction of the first eigentriple group, and other color lines indicate the direction of the remaining eigentriple groups.

**Figure 5 entropy-23-01644-f005:**
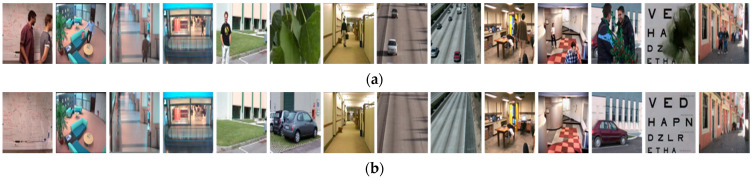
Fourteen different image sequences in the SBI dataset, namely *Board*, *Candela_m1.10*, *CAVIAR1*, *CAVIAR2*, *CaVignal*, *Foliage*, *Hall&Monitor*, *HighwayI*, *HighwayII*, *HumanBody2*, *IBMtest2*, *People&Foliage*, *Snellen*, and *Toscana* (from left to right): (**a**) the frames of image sequences and (**b**) the ground truth background images of image sequences, respectively.

**Figure 6 entropy-23-01644-f006:**
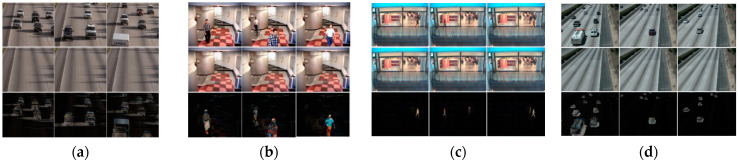
The background (second row) and foreground (third row) images corresponding to the original frames (first row) obtained by using our proposed method in different video sequences in the SBI dataset: (**a**) *HighwayI* sequence, (**b**) *IBMtest2* sequence, (**c**) *CAVIAR2* sequence, and (**d**) *HighwayII* sequence.

**Figure 7 entropy-23-01644-f007:**
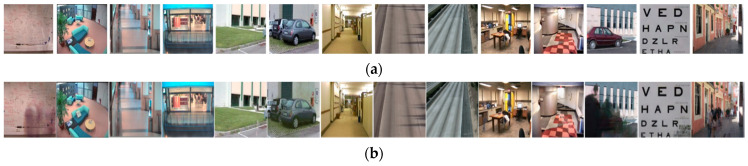
The ground truth background images and obtained background images by using different methods of 14 different image sequences in the SBI dataset, namely *Board*, *Candela_m1.10*, *CAVIAR1*, *CAVIAR2*, *CaVignal*, *Foliage*, *Hall&Monitor*, *HighwayI*, *HighwayII*, *HumanBody2*, *IBMtest2*, *People&Foliage*, *Snellen*, and *Toscana* (from left to right): (**a**) ground truth background images, (**b**) BEWiS, (**c**) median, (**d**) RPCA, (**e**) DMD, (**f**) NMF, and (**g**) BISSA.

**Table 1 entropy-23-01644-t001:** Evaluation metrics.

Eval. Met.	Name	Range of Value	Preference
SSIM [41]	Structural similarity index	[0–1]	higher
FSIM [42]	Feature similarity index for image quality assessment	[0–1]	higher
PSNR [43]	Peak-signal-to-noise ratio	[0–infinity]	higher
AGE [44]	Average gray-level error	[0–255]	lower
pEPs [44]	Percentage of error pixels	[0–1]	lower

**Table 2 entropy-23-01644-t002:** Average results of the various methods in the SBI database.

	Method	SSIM↑	FSIM↑	AGE↓	pEPs↓	PSNR↑
*Board*	BISSA	**0.73**	**0.83**	30.00	0.53	**16.5**
	RPCA	0.63	0.77	40.46	0.64	13.4
	NMF	0.66	0.83	34.88	0.60	15.2
	DMD	0.59	0.74	39.29	0.38	12.3
	BEWiS	0.72	0.82	**18.66**	0.28	16.5
	Median	0.71	0.80	21.74	**0.25**	16.4
*Candela_m1.10*	BISSA	**0.95**	0.93	6.12	**0.03**	25.76
	RPCA	0.94	0.94	4.92	0.03	27.2
	NMF	0.94	0.95	4.64	0.04	27.6
	DMD	0.94	0.94	4.63	0.04	24.8
	BEWiS	0.95	**0.96**	**3.67**	0.03	**28.66**
	Median	0.92	0.93	5.32	0.04	25.78
*CAVIAR1*	BISSA	0.96	**0.97**	6.36	**0.07**	**27.8**
	RPCA	0.91	0.94	12.64	0.12	21.3
	NMF	0.92	0.96	16.70	0.27	22.1
	DMD	-	-	131.85	0.93	-
	BEWiS	**0.97**	0.97	**3.85**	0.45	27.27
	Median	0.90	0.93	9.18	0.08	17.8
*CAVIAR2*	BISSA	0.95	**0.99**	3.20	**0.01**	32.9
	RPCA	0.96	0.98	12.87	0.08	24.8
	NMF	0.96	0.98	13.84	0.11	24.5
	DMD	0.97	0.97	2.49	0.01	30.1
	BEWiS	**0.98**	0.99	**0.78**	0.04	**47.61**
	Median	0.96	0.96	3.40	0.02	25.08
*CaVignal*	BISSA	0.88	0.88	**12.14**	**0.10**	**20.04**
	RPCA	0.74	0.79	26.78	0.36	15.3
	NMF	0.85	0.90	14.89	0.14	19.1
	DMD	0.77	0.84	14.5	0.14	16.5
	BEWiS	**0.96**	**0.98**	12.76	0.10	20.01
	Median	0.83	0.87	12.9	0.10	16.90
*Foliage*	BISSA	0.57	0.72	36.46	0.70	**15.63**
	RPCA	0.57	0.74	41.81	0.56	14.1
	NMF	0.69	0.81	35.96	0.60	15.4
	DMD	0.34	0.63	50.95	0.64	11.6
	BEWiS	**0.87**	**0.91**	**11.8**	**0.17**	15.38
	Median	0.60	0.72	32.30	0.54	23.75
*Hall_monitor*	BISSA	**0.94**	0.93	6.70	**0.03**	**28.3**
	RPCA	0.91	0.94	8.06	0.06	25.1
	NMF	0.93	**0.95**	4.68	0.03	28.1
	DMD	0.89	0.93	6.03	0.04	23.2
	BEWiS	0.92	0.93	3.62	1.43	27.17
	Median	0.90	0.93	**2.7**	0.99	26.46
*HighwayI*	BISSA	**0.95**	**0.95**	7.7	**0.02**	29.08
	RPCA	0.83	0.90	46.68	0.94	14.3
	NMF	0.85	0.92	42.83	0.98	15.1
	DMD	0.66	0.76	18.39	0.29	18.9
	BEWiS	0.94	0.95	2.10	0.46	**54.49**
	Median	0.89	0.93	**1.42**	0.15	40.14
*HighwayII*	BISSA	**0.94**	**0.97**	4.5	**0.003**	33.32
	RPCA	0.93	0.96	4.30	0.01	30.5
	NMF	0.94	0.97	3.57	0.005	33.3
	DMD	0.81	0.89	9.76	0.11	22.4
	BEWiS	0.94	0.96	**2.19**	0.41	34.62
	Median	0.91	0.91	1.72	0.31	**34.66**
*HumanBody2*	BISSA	**0.95**	0.96	9.71	0.12	24.22
	RPCA	0.92	0.94	9.51	0.08	22.5
	NMF	0.95	0.96	8.05	0.08	25.9
	DMD	0.85	0.89	13.0	0.13	18.8
	BEWiS	0.95	**0.98**	**4.26**	1.50	27.97
	Median	0.95	0.97	4.55	**0.01**	**31.96**
*People&Foliage*	BISSA	**0.74**	**0.84**	8.58	0.68	**14.02**
	RPCA	0.62	0.77	7.93	**0.06**	12.2
	NMF	0.67	0.82	**7.22**	0.07	13.0
	DMD	0.46	0.69	10.1	0.07	10.2
	BEWiS	0.66	0.78	34.57	0.40	12.45
	Median	0.66	0.78	31.36	0.38	13.60
*IBMtest2*	BISSA	**0.95**	**0.97**	**41.65**	**0.11**	25.5
	RPCA	0.91	0.94	41.76	0.47	**27.02**
	NMF	0.90	0.94	42.48	0.61	26.8
	DMD	0.88	0.92	53.96	0.57	21.4
	BEWiS	0.94	0.96	43.98	1.50	25.65
	Median	0.86	0.91	48.91	0.15	22.14
*Snellen*	BISSA	0.77	**0.87**	53	0.83	13
	RPCA	0.70	0.82	50.74	0.82	12.9
	NMF	0.76	0.86	**39.92**	0.75	**15.0**
	DMD	0.57	0.76	61.4	0.73	10.9
	BEWiS	**0.76**	0.80	**54.63**	**0.52**	**25.75**
	Median	0.69	0.72	62.20	0.62	13.65
*Toscana*	BISSA	**0.87**	**0.93**	16.18	0.27	**21.23**
	RPCA	-	-	-	-	-
	NMF	0.30	0.77	70.03	0.94	10.3
	DMD	0.76	0.86	22.58	0.28	16.82
	BEWiS	0.85	0.92	10.37	**0.12**	20.87
	Median	0.86	0.94	**8.71**	0.13	20.67

## Data Availability

Not applicable.

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
