# Peer review of "Singular Spectrum Analysis for Background Initialization with Spatio-Temporal RGB Color Channel Data"

_entropy, 2021, doi:10.3390/e23121644_

Round 1
Reviewer 1 Report
This (improved) revised version can be accepted as it is.
Author Response
We would like to express our appreciation for the efforts the Reviewer made to improve the paper’s quality.
Reviewer 2 Report
The authors have improved the previous version of their paper regarding the structure, quality of explanations, and results evaluation.
The experiments described in the paper using different methods with 14 different image sequences in SBI dataset prove that the method proposed by the authors produces better results than the other methods.
However, there are still some editing and English expression mistakes. For example:
- we propose a novel method belong to the low-rank/sparse data separation methods
- the paper concludes in Section V with conclusions
- Embedding: construction the trajectory matrices
- So, the convergence criterion is not met in RPCA, lead to can’t compute the matrix contains the weighting of each pixel in the training data.
Author Response
According to the reviewer’s comments, we have revised the paper.
Thank you so much.
